# The discovery of a novel series of compounds with single-dose efficacy against juvenile and adult *Schistosoma species*

J. Mark F. Gardner[1]*, Nuha R. Mansour[2], Andrew S. Bell[1], Helena Helmby[2], Quentin Bickle[2]

1 Salvensis, Sandwich, Kent, United Kingdom, 2 Department for Infection Biology, London School of Hygiene and Tropical Medicine, London, United Kingdom

* mark.gardner@salvensis.org

## Abstract

Treatment and control of schistosomiasis depends on a single drug, praziquantel, but this is not ideal for several reasons including lack of potency against the juvenile stage of the parasite, dose size, and risk of resistance. We have optimised the properties of a series of compounds we discovered through high throughput screening and have designed candidates for clinical development. The best compounds demonstrate clearance of both juvenile and adult *S. mansoni* worms in a mouse model of infection from a single oral dose of < 10 mg/kg. Several compounds in the series are predicted to treat schistosomiasis in humans across a range of species with a single oral dose of less than 5 mg/kg.

## Author summary

Schistosomiasis (also known as Bilharzia) is a severe disease with WHO estimates suggesting that more than 229 million people require preventative treatment. It is caused by parasitic worms which infect through the skin and, when mature, pair up in different parts of the body according to species, and produce eggs leading to significant adverse health impacts including death. Treatment is reliant on one drug, praziquantel, which is effective against adult worms, but has some disadvantages. Praziquantel is not effective against juvenile worms and is therefore unable to prevent the disease. Also, a large dose is required, and furthermore, reliance on one drug risks the development of drug resistance. This research describes the discovery of a new series of potent compounds, unrelated to praziquantel, which kill both the adult and juvenile parasitic worms *in vitro*, and in mice, following a low single oral dose. Predictions suggest that several of the compounds should be capable of curing infection in people with a single oral dose approximately 10-fold smaller than praziquantel.

## Introduction

Schistosomiasis is a severe disease caused by parasitic worms. 229 million people require treatment [1] and it accounts for up to 300,000 deaths [2,3] and 70 million disability-adjusted life

**Data Availability Statement:** The vast majority of the data is available within the manuscript and supporting information. Data is also available at https://chembl.gitbook.io/chembl-ntd/downloads/

deposited-set-25-schistosoma-dataset-1st-july-2021 and https://chembl.gitbook.io/chembl-ntd/downloads/deposited-set-25-schistosoma-dataset-1st-july-2021

**Funding:** Work was funded by the Medical Research Council, United Kingdom, grant award number MR/ K025430/1 made to QB (https://mrc.ukri.org/). The funders had no role in study design, data collection and analysis, decision to publish or preparation of the manuscript.

**Competing interests:** I have read the journal's policy and the authors of this manuscript have the following competing interests: Since the project work was completed JMFG has taken on a small consultancy with Merck KGaA who are now the assignee of the patents covering this work.

years [4] annually mainly in sub-Saharan Africa. Humans are infected percutaneously by the cercarial larval stage shed from aquatic snails and thereafter the worms remain in the human blood stream migrating from the skin, through the lungs and maturing in the liver. The adult worms then migrate to the mesenteric veins or the vesical plexus of the bladder, depending on species, where they lay eggs which cause inflammation and fibrosis in various organs.

Praziquantel (PZQ) is the only drug recommended for treatment of schistosomiasis and its increasingly widespread use in mass chemotherapy campaigns means it is the mainstay of control of this infection [5]. PZQ has proved to be generally safe and effective using a single oral dose [6,7], however reliance on a single drug has led to concerns over potential development of drug resistance especially as mass drug administration campaigns will increase drug pressure [8,9]. Although PZQ is active against adult worms of all the medically important *Schistosoma* species [10], it is relatively ineffective against the juvenile stages both *in vivo* and *in vitro* [11,12]. Finally, while PZQ is effective at reducing infection intensity, it achieves modest cure rates in children of 73.6% (*S. haematobium*), 76.4% (*S. mansoni*) and 95.3% (*S. japonicum*) at the WHO recommended dose of 40 mg/kg [13]. Consequently, there is a need for new anti-schistosomicides, which has led to renewed interest in research into drug discovery using both phenotypic and target-based approaches. These include development and application of whole organism screens for compound testing [14], target-based drug discovery [15] and identification of putative molecular targets by analysis of the annotated schistosome genome sequences [16]. A recent comprehensive review of compounds with schistosomicidal activity [17] demonstrates the relative lack of promising series in the area.

Considering the needs of the target population and profile of praziquantel, in this work we sought a target candidate profile including the following properties: balanced activity against both juvenile and adult worms of all three clinically relevant species; curative from a single oral dose of less than 10 mg/kg in human; simple and cheap to synthesise and formulate; and displaying a large therapeutic index. We have previously described the discovery of several hits from a high-throughput screen (HTS) [18] based on use of the larval, schistosomula stage for primary screening of large compound libraries [19]. Here we describe the medicinal chemistry carried out on the hits and the optimization of two structurally related hits LSHTM-1507 and LSHTM-1945 (Table 1) to promising candidates for clinical development, which may subsequently prove to be useful in the treatment of schistosomiasis.

## Materials and methods

### Ethics statement & animals

Experimentation was carried out using the NC3Rs and ARRIVE guidelines under the United Kingdom's Animals (Scientific Procedures) Act 1986 (under project licences 60/4456 and PD76183B4) with approval from the London School of Hygiene and Tropical Medicine Animal Ethics committee. CD1 (Swiss albino) mice (aged 5–6 weeks) were bred on site using SPF conditions with access to food and water *ab libitum*.

### Compound synthesis and characterization

All new compounds were prepared by standard methods by TCG Life Sciences [20]. The general scheme for the synthesis of final compounds involved the synthesis of a substituted 5,6 aromatic core ring system (Fig 1). This was decorated with bromine or iodine in the required substitution pattern. Aromatic substituents were added to the 5,6 core using Suzuki chemistry making use of differential reactivity of the halides (usually iodine and bromine). The second halide was occasionally added after the first Suzuki reaction. A detailed route for the synthesis of 10 g of LSHTM-3642 is provided in the supporting information (S1 Text). All final

**Table 1. Structure and activity of hits and early analogues.**

| Compound number | Core | R₁ | R₂ | R₃ | in vitro assays* | | | | clogD† |
|---|---|---|---|---|---|---|---|---|---|
| | | | | | Adult *S. mansoni* EC$_{50}$ (nM) | Juvenile *S. mansoni* EC$_{50}$ (nM) | Cytotoxicity TC50 (nM) | Cytotox cell line | |
| LSHTM-1945 | a | 4-methoxyphenyl | isopropoxy | H | 4,900 | 5,700 | 40,100 | MRC5 | 3.9 |
| LSHTM-1507 | b | 3,4-dimethoxyphenyl | isopropyl | H | 1,600 | 4,000 | 18,600 | MRC5 | 4.2 |
| 1 | a | 4-methoxyphenyl | isopropyl | H | 1,850 | 2,430 | 9,040 | hepG2 | 4.2 |
| 2 | a | 4-methoxyphenyl | H | isopropyl | | >25,000 | | | |
| 3 | a | 4-hydroxyphenyl | isopropoxy | H | 4,430 | 5,310 | | | 3.6 |
| 4 | a | 1H-indazol-5-yl | isopropoxy | H | 7,190 | 7,500 | 22,200 | hepG2 | 3.7 |
| 5 | a | 3-trifluoromethoxyphenyl | isopropyl | H | | >25,000 | | | 5.1 |
| 6 | a | 4-hydroxyphenyl | isopropyl | H | 1,340 | 1,500 | 30,000 | MRC5 | 3.9 |
| 7 | a | 3-hydroxy, 4-methoxyphenyl | isopropyl | H | 263 | 486 | 18,600 | MRC5 | 3.9 |
| 8 | b | 2-fluoro, 4-hydroxyphenyl | isopropyl | H | 262 | 636 | 11,100 | MRC5 | 4.0 |
| 9 | b | 4-hydroxyphenyl | isopropyl | H | | >25,000 | | | 3.9 |
| 10 | b | 3-hydroxy, 4-methoxyphenyl | isopropyl | H | | 9,740 | | | 3.9 |
| 11^ | c | 4-(methylsulfonyl) phenyl | 3-(methylsulfonyl) phenyl | H | | 25,000 | 11,900 | hepG2 | 1.9 |
| 12 | c | 3-hydroxy, 4-methoxyphenyl | isopropyl | H | 5,590 | 8,260 | 9,240 | hepG2 | 3.9 |
| 13 | c | 2-fluoro, 4-hydroxyphenyl | trifluoromethyl | H | >25,000 | >25,000 | >50,000 | MRC5 | 3.9 |

* incubation times for *S. mansoni* and MRC5 assays were 120 h, for hepG2 72 h.

† clogD is calculated using SlogP-1 as this gives a reasonable estimate of experimental logD for this series. SlogP was calculated using RDKit in KNIME [25].

^ Compound 35 [26].

compounds were characterized as a minimum by analytical HPLC-MS and by ${}^1$H NMR. The synthesis (and biological activity) of compounds in this paper is described in patent application WO2018130853 [21]. A look-up table is provided in the supporting information (S1 Table) to more speedily allow the reader to check the synthetic details of particular examples in the patent.

## Parasite maintenance, preparation and assay methods

The Puerto Rican strain of *S. mansoni* was maintained in *Biomphalaria glabrata* and CD1 mice. *Oncomelania hupensis* (*hupensis*) infected with S. *japonicum* (Chinese strain) and *Bulinus truncatus* (*truncatus*) infected with Egyptian strain *S. haematobium* were obtained from BEI Resources, NIAID, NIH. Infection of mice for worm recovery was as previously described [18] except that, whereas cercariae of *S. mansoni* and *S. haematobium* were obtained by shedding from infected snails under lights, cercariae of *S. japonicum* were obtained by gently

**Fig 1. General scheme for the synthesis of final compounds.** X = I (or Br), Y = Br (or I).

crushing the snails. Furthermore, adult *S. haematobium* were recovered after 12 weeks following mouse infection.

*In vitro* efficacy was carried out as previously described [18]. In brief, worms were recovered from infected mice by portal perfusion 3 weeks (for juveniles) or 6 weeks (for adults) post-infection using warm perfusion medium (Dulbecco's Modified Eagle's Medium [DMEM], 2mM L-glutamine, 100 Units/ml penicillin, 100μg/ml streptomycin, 20mM Hepes, 10 Units/ml heparin [Sigma, UK]). Worms were washed free of red blood cells using the perfusion medium, and finally suspended in complete medium (cDMEM: DMEM, 2mM L-glutamine, 100 Units/ml penicillin, 100μg/ml streptomycin, 10% FCS). The juvenile assays were carried out in 96-well plates, each well containing 6–8 worms in 200 μL cDMEM. The adult worm assay was in 24-well plates, each well containing 3 worm pairs in 1mL cDMEM. Both the juvenile and adult worm cultures were kept at 37˚C, in an atmosphere of 5% $CO_2$. Drug effects were determined on day 5 following drug treatment by assessing the viability of individual worms as described in [14].

Prior to *in vivo* efficacy experiments, a single animal was given a dose of drug (using the same formulation) that was twice as large as that planned for use in the efficacy experiments and was observed periodically for adverse clinical signs for 24h by trained veterinarians. Adverse clinical signs checked include postural defect, vocalization, oculo-nasal discharge, hypersalivation, change in activity, abnormal gait, respiration and pulse, piloerection, loss of pupillary reflex, eyelid reflex, chromodacryorrhea, reaction to handling, tremor, convulsions, prostration, diarrhoea, cachexia, emaciation. No such effects were seen with any of the compounds tested. *In vivo* efficacy was assessed in female CD1 mice using methods based on those previously described [14]. Mice were infected subcutaneously with 150 cercariae. Test compounds were suspended in 7% Tween-80 / 3% Ethanol / double distilled water and drug dispersal was facilitated by vortexing and using a sonicating water bath (Formulation "Aqueous"). Alternatively, drugs were first dissolved in DMSO, then diluted with corn oil to give a 1:19 DMSO:corn oil solution/suspension of test compound (Formulation "Corn oil"). Dosing was by oral gavage. Mice in the negative control groups were given the drug suspension vehicle only. Positive controls for treatment of adult, day 42 old, infections were treated with PZQ and those for juvenile, day 21 old, infections with artemether, which unlike PZQ is effective against the juvenile worm infections in mice [22]. During and following treatment, mice were closely observed for any ill effects by staff experienced in animal handling. This involved individuals who were blinded to the treatment groups. Effects they were trained to look for included hunching, immobility, piloerection, subdued behaviour, diarrhoea and loss of body condition. None were seen. Worms were recovered by portal perfusion with citrate saline 8–14 days post treatment and counted. Differences between groups relative to the non-drug treated controls were assessed for statistical significance by Student's unpaired t-test using GraphPad Prism Software. P values are presented as follows:- ns–P > 0.05 (not significant), * P ≤ 0.05, ** P ≤ 0.01, *** P ≤ 0.001, **** P ≤ 0.0001.

## Cytotoxicity assays

Activity against mammalian cells was determined using either of two standard assays. The first of these used MRC-5 cells with Alamar blue staining as previously described [18]. The second used HepG2 cells monitored using 3-(4,5-dimethylthiazol-2-yl)-2,5-diphenyltetrazolium bromide (MTT) as previously described [23].

## Pharmacokinetics

Routine pharmacokinetics was carried out as previously described [24] with the following modifications: Swiss albino male mice of average weight 30 g with three animals per

experiment. The final vehicle was 10% DMSO, 90% 50 mM $Na_2HPO_4$ with 0.5% Tween-80. Compounds were dosed singly (at a variety of doses as noted in the text) or more commonly in mixtures of 5 at 1 mg/kg per compound iv into the lateral tail vein or 2.5 mg/kg per compound by oral gavage. ~30 µl blood samples were taken by venepuncture, into heparinized capillary tubes and subsequently transferred into 0.5 ml micro-centrifuge tubes. Collection time points were 5 min (iv only), 15 min, 30 min, 1 h, 2 h, 4 h, 8 h & 24 h. All blood samples were processed for plasma by centrifugation at 4,000 rpm for 10 min at 4˚C within half an hour of collection. Plasma samples were stored at -20˚C until bioanalysis.

For pharmacokinetic analysis during the *in vivo* efficacy experiments individual animals were only sampled once. Samples were taken from one animal in each experimental group at each of 1, 4, 8 h and samples from two animals in each group were taken at 24 h.

## Bioanalysis of plasma samples

Prior to LC-MS analysis, all samples were extracted using protein precipitation using acetonitrile as the extraction solvent. For each batch of samples, a calibration line spanning an appropriate concentration range was created by spiking the compound of interest into blank matrix. The calibration line included 10 evenly spaced points. To ensure reproducibility, QC samples were included at a minimum of 3 levels (low, mid and high concentrations). Where necessary, additional dilution QCs were included to validate the dilution process. Within each batch at least 3 study samples were assayed pre and post dilution to assess whether there was any impact from the presence of dosing vehicle in the samples. A linear regression was performed on peak area response ratios (analyte:IS) and the results used to back-calculate analyte concentrations. A batch of samples were considered acceptable if 75% of the calibration samples and 66% of the standard QC samples were within 20% of nominal. The linear range of the assay was defined by the calibration line ±20% i.e. the maximum reportable concentration was 120% of the highest concentration calibration sample that passed acceptance criteria. A metric plot of the IS response was reviewed to ensure the integrity of the study sample data.

## Plasma protein binding in mouse or human plasma measured at 10 uM

Each experiment was performed in duplicate for the compound being tested. A working solution of compound was prepared from 10 µL of 10 mM solution in DMSO + 190 µL 40:60 MeCN:Water. A Total solution was prepared from 12 µL of working solution + 588 µL plasma (compound conc: 10 µM, DMSO: 0.1%, ACN:0.76%). In an equilibrium dialysis chamber, Total solution (200 µL) was added to the plasma chamber and PBS pH 7.4 (350 µL) was added to the buffer chamber. The chamber was then incubated at 37˚C for 6 h with constant shaking at 400 rpm (Thermomixer comfort, Eppendorf). Aliquots were removed from both plasma and buffer chambers and the following test samples prepared: Blank: blank Plasma (20 µL) + blank buffer (60 µL), Buffer: blank plasma (20 µL) + test buffer (60 µL), Plasma: test plasma (20 µL) + blank buffer (60 µL), Total: total sample (20 µL) + blank buffer (60 µL). Ice-cold acetonitrile (160 µL), containing a suitable standard was added to all samples, they were shaken vigorously and centrifuged at 4,000 rpm for 20 min at 15˚C. Supernatant (110 µL) from each sample was diluted with water (110 µL) and the sample analysed by LC-MS/MS.

Calculations:

% Free = 100 × (AREA in buffer × dilution factor)/(AREA in Plasma × dilution factor)

% Bound = (100 - % Free)

% Recovery = 100 ×[(Avg AREA in buffer × dilution factor) + (Avg AREA in Plasma × dilution factor)] / (Avg AREA in Total × dilution factor)

## LogD

Pre-saturated solvents were prepared as follows: Octanol (100 mL) and PBS pH 7.4 (100 mL) were mixed by rotospin at room temperature for 24 h, then centrifuged at 2,000 rpm for 5 mins. The layers were separated into aqueous and organic phases. Compound (10 mM, 4.5 μL) was transferred to a 96 deep well assay plate, octanol (pre-saturated with PBS) (300 μL) was dispensed to each well and the plate shaken for 2 min using Thermomixer. PBS (pre-saturated with Octanol) (300 μL) was added, the plate sealed, and shaken at 850 rpm for 2 h at 25˚C using Thermomixer. The plate was allowed to stand for 30 min and centrifuged at 2,000 rpm for 10 mins at 25˚C. 100 μL of the octanol layer was transferred to a 96 well plate and the remaining octanol discarded. The plate was centrifuged at 2,000 rpm for 5 min at 25˚C. 100 μL of buffer layer was transferred to a 96 well plate. Diluents used were:- Diluent 1: 10:90 Water: MeOH. Diluent 2: 10:90 Water:MeOH containing system suitability standard (0.2 μg/mL). *Processing the Octanol Layer*: Step-1: 1:20 dilution: octanol layer (20 μL) + Diluent 1 (380 μL). Step-2: 1:50 dilution: Step-1 (60 μL) + Diluent 1 (90 μL). Step-3: 1:1,000 dilution: Step-2 (20 μL) + Diluent 2 (380 μL). *Processing the Buffer Layer*:Dilution1: 1:20 dilution: buffer layer (20 μL) + Diluent 2 (380 μL). The samples were analysed using LC-MS/MS.

LogD is expressed as Log[AUC in octanol layer * Relative dilution factor / AUC in buffer layer]

## Metabolic stability study using mouse or human hepatoyctes

100 μM compound stock solution was prepared from 5 μL of 4 mM stock in DMSO + 195 μL MeCN. Working solution (2 μM compound) was prepared using 10 μL of 100 μM stock + 490 μL buffer pH 7.4 (KHB supplemented with $CaCl_2$, $NaHCO_3$, HEPES, fructose, glycine; pH 7.4).

Working solution (25 μL, 2 μM) was added in duplicate to wells in a 0.5 mL per well, 96 well plate which was preincubated to reach 37˚C. Hepatocyte suspension (25 μL) was added to each plate, and the plate incubated (37˚C, 95% relative humidity with 5% $CO_2$ supply) under gentle mixing (Thermomixer @300 rpm). After designated time periods, ice-cold MeCN (250 μL) was added with a suitable standard. For T = 0 min, ice-cold MeCN (250 μL) with suitable standard was added to the working solution (2 μM) and hepatocyte suspension (25 μL) added. The plates for each time point were shaken, sonicated for 5 mins and kept at 4˚C until the final plate was processed. The plates were centrifuged at 4,000 rpm for 20 min, supernatant (110 μL) was mixed with water (110 μL) and the samples analysed using LC-MS/MS.

% Remaining at time point t = 100 × [(AUC at time point t) / (AUC at T = 0)]

## Broad selectivity panel

Selectivity was assessed at Cerep (Celle l' Evescault, France), as described in the Cerep catalog, against a panel of *in vitro* radioligand receptor binding assays with 38 different receptors, channels, and transporters selected as targets to avoid for safety reasons (beta 1, beta 2, NMDA, MAO-A, 5-HT1B, CCK1 (CCKA), BZD (central), delta (DOP), A2A, CB1, CB2, D1, D2S, ETA, H1, H2, M1, M2, M3, 5-HT1A, 5-HT2A, 5-HT2B, 5-HT3, GR, AR, V1a, kappa (KOP), mu (MOP), alpha 1A, alpha 2A, Ca2+ channel, Potassium Channel, Kv channel, Na + channel, N neuronal alpha 4 beta 2, norepinephrine transporter, dopamine transporter, 5-HT transporter). The specific ligand binding to the receptors is defined as the difference between the total binding, and the nonspecific binding determined in the presence of an excess of unlabelled ligand. The results are expressed as a percent of control specific binding and as the mean percent inhibition of control specific binding obtained in the presence of 1 μM and 10 μM compound.

Kinase panel activity: Activity of compounds at 1 µM was assessed against a panel of 50 kinase enzymes (MKK1, JNK1, p38a MAPK, RSK1, PDK1, PKBa, SGK1, S6K1, PKA, ROCK 2, PRK2, PKCa, PKD1, MSK1, CAMKKb, CAMK1, SmMLCK, CHK2, GSK3b, PLK1, Aurora B, LKB1, AMPK (hum), MARK3, CK1δ, CK2, DYRK1A, NEK6, TBK1, PIM1, SRPK1, EF2K, HIPK2, PAK4, MST2, MLK3, TAK1, IRAK4, RIPK2, TTK, Src, Lck, BTK, JAK3, SYK, EPH-A2, HER4, IGF-1R, TrkA, VEG-FR) carried out by the MRC Protein Phosphorylation and Ubiquitinylation Unit at the University of Dundee, Dundee, DD1 5EH, UK.

Ion channel activity. Activity of compounds against ion channels was assessed by Metrion biosciences, Cambridge, CB21 6AD, UK in dose-response experiments (n = 2) up to 10 µM compound concentration with standards & equipment as listed for each ion channel (hERG–verapamil—QPatch, Nav1.5 –amitryptiline—QPatch, Kv1.5 –transDSC—QPatch, Cav1.2 –nimodipine—Flex).

## Results

### Potency improvement

All the hits from the high throughput screen were relatively weak ($EC_{50}$s 1–10 µM) with narrow margins over cytotoxicity in some cases (Table 3 in our previous paper [18]). The key priority to demonstrate that a series could be taken forward was to improve potency in the primary *in vitro* assay and to increase selectivity with respect to cytotoxicity in a human cell line. For the first exploratory steps in each series, we selected the juvenile (3 week old) *S. mansoni* worm assay as our primary assay because one of our key aims was to identify compounds with better activity against the juvenile worms than PZQ. Most of the actives were also tested in the *S. mansoni* adult worm assay and mammalian cell line assays to check for cytotoxicity.

Initially we chose to explore the potential for potency improvement of each of the seven hits, however, it was only possible to increase the ratio of potency to cytotoxicity in the series exemplified by LSHTM-1507 and LSHTM-1945 (Table 1). To explore the potential around these hits in a cost-effective manner, we made use of parallel medicinal chemistry; 193 compounds were synthesised and tested including 94 imidazopyrazines and 72 triazolopyridines. We observed that of the two cores represented by LSHTM-1507 and LSHTM-1945, the imidazopyrazine conferred greater activity. The imidazopyrazine compound library work served to demonstrate that it was only rarely possible to introduce significant polarity at positions $R_1$ or R and maintain potency (Fig 2). The only significant polarity that retained potency was the replacement of methoxy-phenyl in position $R_1$ with indazole (compound 4) or phenol (compound 3). The substitution R on the phenyl ring needed to be meta and lipophilic. This is illustrated in Table 1 (and Fig 3) by comparison of the pairs compound 6 vs compound 9 and compound 7 vs compound 10.

We were also aware of the potentially related imidazopyridazine structure (11, compound 35 in Le Manach et al [26]) disclosed as a potent anti-plasmodial agent with good activity both *in vitro* ($IC_{50}$ *vs P. falciparum* K1 6 nM, NF54 7 nM) and *in vivo*. We investigated the potential for utilising the precedented imidazopyridazine core (c in Table 1) with the synthesis of compounds 11, 12 and 13. However, not only did the potent anti-plasmodial agent compound 11 have almost no activity against *S. mansoni* (at least 1,000 fold weaker), substitutions that have good activity against S. *mansoni* with the imidazopyrazine core (a, compounds 7 Table 1, and 15 Table 2) are considerably weaker against S. *mansoni* when attached to the imidazopyridazine core (c). Given the two cores are isosteric, the difference of SAR is remarkable.

All of the active compounds at this time included either a hydrogen bond donor (phenol or indazole) or a methoxy group at the 4-position of side chain $R_1$ and we decided to probe the SAR of this feature further. We postulated that the methoxy could be cleaved to form the

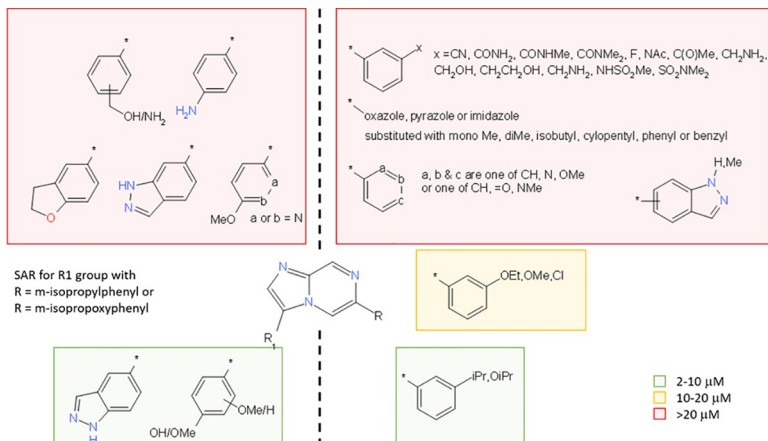

**Fig 2. Compound library SAR.** Illustration of the structure activity relationship derived from an exploratory set of compounds made as a combinatorial compound library and tested *in vitro* against *S. mansoni* juvenile worms for 5 days. Colour coded to show $EC_{50}$ values.

phenol by enzymatic activity of the worm, and that the activities of the methoxy species could be due in fact to the phenol metabolite. Metabolic activation by the parasite is precedented; for example, it is the mechanism by which oxamniquine is converted into the active species [20]. Replacement of the methoxy of compound 1 ($EC_{50}$ 2.4 µM) with trifluoromethyl (compound 5) results in a complete loss of activity ($EC_{50}$ > 25 µM), whereas the phenol (compound 6) is similarly potent ($EC_{50}$ 1.5 µM). This is by no means proof of the metabolic cleavage hypothesis but is at least an interesting SAR point. Other phenols displayed a range of activities with 4-methoxy 3-phenol (compound 7) and 2-fluoro 4-phenol (compound 8) giving us our first compounds with a submicromolar $EC_{50}$.

During this time we also explored the $R_2$ position. We were not able to find groups that improved potency, or indeed polar groups compatible with potency. However, the substitution pattern is important; for example, 3-substituted isopropyl (compound 1) has an $EC_{50}$ of 2.4 µM whereas the 4-substituted isopropyl (compound 2) is inactive ($EC_{50}$ > 25 µM).

After completion of the initial optimisation phase, with selectivity over *in vitro* cytotoxicity greater than 40 fold, we began to explore the effect of substitution at other positions around the core (Table 2 and Fig 4). We hypothesised that the positions adjacent to the heterocyclic N atoms ($R_1$, $R_4$, $R_5$) might afford opportunities to improve potency and/or metabolic stability so focused our efforts there. Substitution at $R_5$ with CN, OMe, SMe or Me resulted in significantly worse potency relative to the H equivalents. However, substitution at $R_4$ with a Me group lead

**Fig 3. Guide to core structures of compounds in Table 1 (letters refer to column "Core").**

**Table 2. Structure activity relationship of more potent and stable compounds.**

| compound number | R$_1$ | R$_2$ | R$_3$ | R$_4$ | Adult S. mansoni EC$_{50}$ (nM) | Juvenile S. mansoni EC$_{50}$ (nM) | Cytotox TC$_{50}$ (nM) | Cytotox cell line | Mouse hepatocyte % remaining at 30/90 min | Human hepatocyte % remaining at 30/90 min | clogD[†] |
|---|---|---|---|---|---|---|---|---|---|---|---|
| | Side chains | | | | in vitro assays* | | | | | | |
| 14 | 4-hydroxyphenyl | isopropyl | H | methyl | 430 | 421 | >11,500 | hepG2 | | | 4.2 |
| 15 | 2-fluoro-4-hydroxyphenyl | trifluoromethyl | H | H | 2,320 | 2,120 | >= 38,300 | MRC5 | 33/- | 31/- | 3.9 |
| 16 | 2-fluoro-4-hydroxyphenyl | trifluoromethyl | H | methyl | 290 | 614 | 18,700 | MRC5 | 32/- | | 4.2 |
| 17 | 2-chloro-4-hydroxyphenyl | trifluoromethyl | H | H | | 4,260 | | | | | 4.4 |
| 18 | 4-hydroxyphenyl | trifluoromethyl | H | methyl | 615 | 1,420 | | | | | 4.1 |
| 19 | 2,6-difluoro-4-hydroxyphenyl | trifluoromethyl | H | methyl | | 172 | 8,400 | hepG2 | 42/- | | 4.4 |
| 20 | 2-fluoro-4-hydroxyphenyl | trifluoromethyl | H | isopropyl | 122 | 79 | 9,000 | hepG2 | 54/- | | 5.1 |
| 21 | 2,6-difluoro-4-hydroxyphenyl | trifluoromethyl | H | isopropyl | 46 | 94 | 7,500 | hepG2 | -/30 | | 5.2 |
| 22 | 3-hydroxy-4-methoxyphenyl | trifluoromethyl | H | isopropyl | 82 | 66 | 11,100 | hepG2 | 36/- | | 4.9 |
| LSHTM-3520 | 2,6-difluoro-4-hydroxyphenyl | trifluoromethyl | F | isopropyl | 46 | 27 | 7,130 | hepG2 | 65/44 | 52/- | 5.3 |
| 23 | 3-hydroxy-4-methoxyphenyl | trifluoromethyl | F | isopropyl | 49 | 177 | | | 55/- | | 5.1 |
| 24 | See Fig 4 | trifluoromethyl | F | trifluoromethyl | 329 | 350 | | | -/56 | | 5.1 |
| 25 | See Fig 4 | trifluoromethyl | F | trifluoromethyl | | 760 | | | | | 5.4 |
| LSHTM-3642 | 4-fluoro-1H-indazol-5-yl (See Fig 4) | trifluoromethyl | F | trifluoromethyl | 27 | 48 | 16,600 | hepG2 | -/90 | -/105 | 5.3 |
| LSHTM-3608 | 4-fluoro-1H-indazol-5-yl | pentafluoroethyl | H | trifluoromethyl | 15.3 | 21.0 | 10,900 | hepG2 | -/88 | -/98 | 5.8 |
| LSHTM-3686 | 4-fluoro-1H-indazol-5-yl | isopropyl | F | cyclopropyl | 5.4 | 4.1 | 15,800 | hepG2 | -/75 | -/75 | 5.2 |
| LSHTM-3645 | 4-fluoro-1H-indazol-5-yl | pentafluoroethyl | F | trifluoromethyl | 9.7 | 10.7 | 11,700 | hepG2 | -/94 | -/105 | 5.9 |
| LSHTM-3661 | 4-fluoro-1H-indazol-5-yl | isopropyl | F | trifluoromethyl | 2.3 | 2.7 | 17,000 | hepG2 | -/78 | -/45 | 5.4 |
| LSHTM-3604 | 4-fluoro-1H-indazol-5-yl | trifluoromethyl | F | isopropyl | 31.5 | 33.2 | 12,000 | hepG2 | 89/80 | 99/98 | 5.4 |
| LSHTM-3690 | 4-fluoro-1H-indazol-5-yl | isopropoxy | F | trifluoromethyl | 9.0 | 11.1 | 9,250 | hepG2 | -/31 | -/79 | 5.0 |

All compounds in Table 2 have R$_5$ = H.

* incubation times for S. mansoni and MRC5 assays were 120 h, for hepG2 72h.

† clogD is calculated using SlogP-1 as this gives a reasonable estimate of experimental logD for this series. SlogP was calculated using RDKit in KNIME [25].

to a modest potency improvement (compound 14) and provided a point for further SAR exploration (Table 2). Further substitution of the R$_4$ position can produce significant potency improvement, with several compounds having an in vitro EC$_{50}$ below 100 nM. Substitution of the phenol or indazole by F ortho to the imidazopyrazine ring results in potency improvement. Comparing entries compounds 16, 18 and 19, or compounds 17 and 15, or compounds 24, 25 and LSHTM-3642 gives a potential rationale for this potency improvement. It could be due to

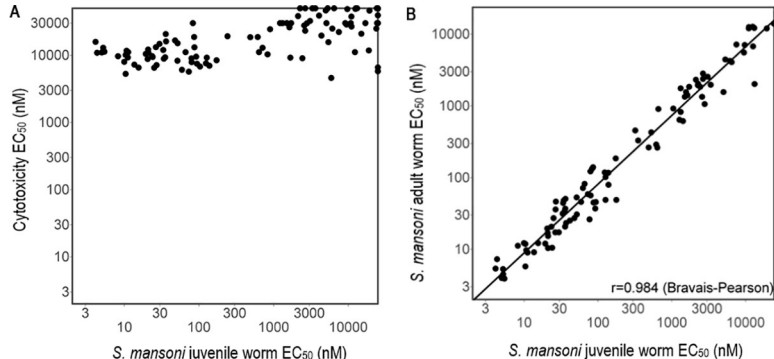

Core structure of all
compounds in Table 2

24          25          LSHTM-3642

Structure of R1 for compounds shown in Table 2

**Fig 4. Guide to structures in Table 2.**

the electron withdrawing nature of F rather than an increased twist of the phenyl and imidazo-pyrazine ring as Cl and Me give smaller potency improvements compared to F.

The further potency improvements demonstrated by several compounds in Table 2 result in a concomitant improvement in selectivity over cytotoxicity since the cytotoxicity has remained at the same level as the original hits (Fig 5A). Fig 5B demonstrates that for this series (unlike PZQ) the *in vitro* potency against juvenile (21 day old) *S. mansoni* worms is the same as that against adult (42 day old) worms.

## Potency against different species of *Schistosoma*

Species difference in the treatment of schistosomiasis is well precedented in the field. Praziquantel has a broad spectrum of activity against adult schistosome worms from the most important species in human disease, *S. mansoni*, *S. haematobium* and *S. japonicum*. However, oxamniquine is effective against *S. mansoni* infections but not *S. haematobium* infections [27]. Initial studies showed our series to be effective *in vitro* against all three species. Tests on later compounds have confirmed potent activity against all three species, though with some evidence that substitution at $R_1$ may have some impact on the ratio of potencies between the two most clinically relevant species *S. mansoni* and *S. haematobium* and that found in SE Asia, *S. japonicum* (Table 3).

**Fig 5. *In vitro* potency against *S. mansoni*.** Both plots include compounds not shown in tables. **A** Potency improvement obtained against *S. mansoni* worms resulted in a very large (>1,000 fold in many cases) ratio between cytotoxicity (as measured against hepG2 or MRC5 cells) and potency. **B** This series has equal potency against both juvenile (21 days old) and adult (42 days old) *S. mansoni* worms. The Bravais-Pearson correlation r value for the fitted line shown is 0.98.

**Table 3. SAR table showing potency of compounds against different clinically relevant *Schistosoma* species *in vitro*.**

| Compound number | *Sm* adult EC$_{50}$ (nM) | *Sh* adult EC$_{50}$ (nM) | Potency ratio *Sh/Sm* | *Sj* adult EC$_{50}$ (nM) | Potency ratio *Sj/Sm* | R$_1$ side chain class |
|---|---|---|---|---|---|---|
| LSHTM-3608 | 15.3 | 9.3 | 0.6 | 95.2 | 6.2 | Indazole |
| LSHTM-3686 | 5.4 | 3.7 | 0.7 | 46.9 | 8.7 | Indazole |
| LSHTM-3642 | 27.3 | 23 | 0.8 | 389 | 14.2 | Indazole |
| LSHTM-3645 | 9.7 | 6.5 | 0.7 | 129 | 13.3 | Indazole |
| LSHTM-3661 | 2.3 | | N/A | 24.8 | 10.9 | Indazole |
| LSHTM-3520 | 46.2 | 17.9 | 0.4 | 73.5 | 1.6 | Phenol |
| 22 | 82 | 61 | 0.7 | 85 | 1.0 | Phenol |

*Sm* = *S. mansoni*, *Sh* = *S. haematobium*, *Sj* = *S. japonicum*. All incubations were for 120 h. Compound structures are shown in Table 2.

## *In vivo* efficacy

The next major objective for the project was to demonstrate activity in an *in vivo* mouse model of *S. mansoni* infection. During the work to improve potency we had been striving to keep the physicochemical parameters (eg logP, MWt, HB donors & acceptors, rotatable bonds, aromatic ring count) in a range consistent with a high probability of good drug-like properties such as solubility, absorption, stability, low clearance. Although the potency improvements made frequently displayed tight SAR requirements, only the core and R$_1$ were compatible with significant polar functionality. Thus, the most potent compounds were now quite lipophilic, and we were not surprised to see evidence for high metabolic clearance *in vitro*. Although potent, compound 22 has low stability in the mouse hepatocyte assay (36% remaining after 30 min incubation) which is borne out by high clearance in mouse PK (iv 0.5 mg/kg, Cl$_{obs}$ 4.8 L/h/kg, Vd$_{ss}$ 5.1 L/kg, t$_{1/2}$ 1.9h, AUC 0.1 h*ug/mL, oral 2.5 mg/kg, bioavailability 8%).

The best compounds in terms of potency and mouse hepatocyte stability were selected for mouse pharmacokinetics prior to efficacy studies. To improve metabolic stability in the context of SAR that restricts us to fairly lipophilic compounds, we reasoned that we would need to block suspected vulnerable positions. We also sought to maintain the rigidity of the molecules as we believed that this might restrict access of the molecule to the correct alignment in active sites of the metabolising enzymes. Replacement of the methoxy-phenol present in compound 22 with monofluoro- (compound 20) or difluorophenol (compound 21) improves metabolic stability (Table 2). Similarly, blocking the 4 position of the trifluoromethylphenyl ring (compound 23) with fluorine also improves metabolic stability. Combining these changes together in LSHTM-3520 results in a potent compound (EC$_{50}$ 27 nM and 46 nM in juvenile and adult worms respectively) with a half-life in mouse hepatocytes (t$_{1/2}$ = 1.25 h) approximately four times that of compound 22. The improvement in hepatocyte stability feeds through into a significant improvement in mouse PK (iv dose 0.5 mg/kg, Cl$_{obs}$ 1.1 L/h/kg, Vd$_{ss}$ 7.1 L/kg, t$_{1/2}$ 7.0h, AUC 0.5 h*µg/mL, oral dose 2.5 mg/kg, bioavailability 50%).

With good potency and pharmacokinetics we felt that LSHTM-3520 should be good enough to demonstrate efficacy *in vivo*. Indeed, LSHTM-3520 was highly effective in reducing the *S. mansoni* worm count in mice infected for 42 days (adult worms) at 200 mg/kg from a single oral dose (Fig 6A).

In order to link the observed efficacy to the low dose PK data, the blood of the treated animals was sampled (once per animal) and the plasma concentrations of parent drug were quantified. Fig 6B demonstrates that the plasma levels observed during the efficacy experiment closely match the levels predicted by the PK experiment. No significant changes are observed in the behaviour of the drug at elevated doses. Although not a safety study, it is worth noting that

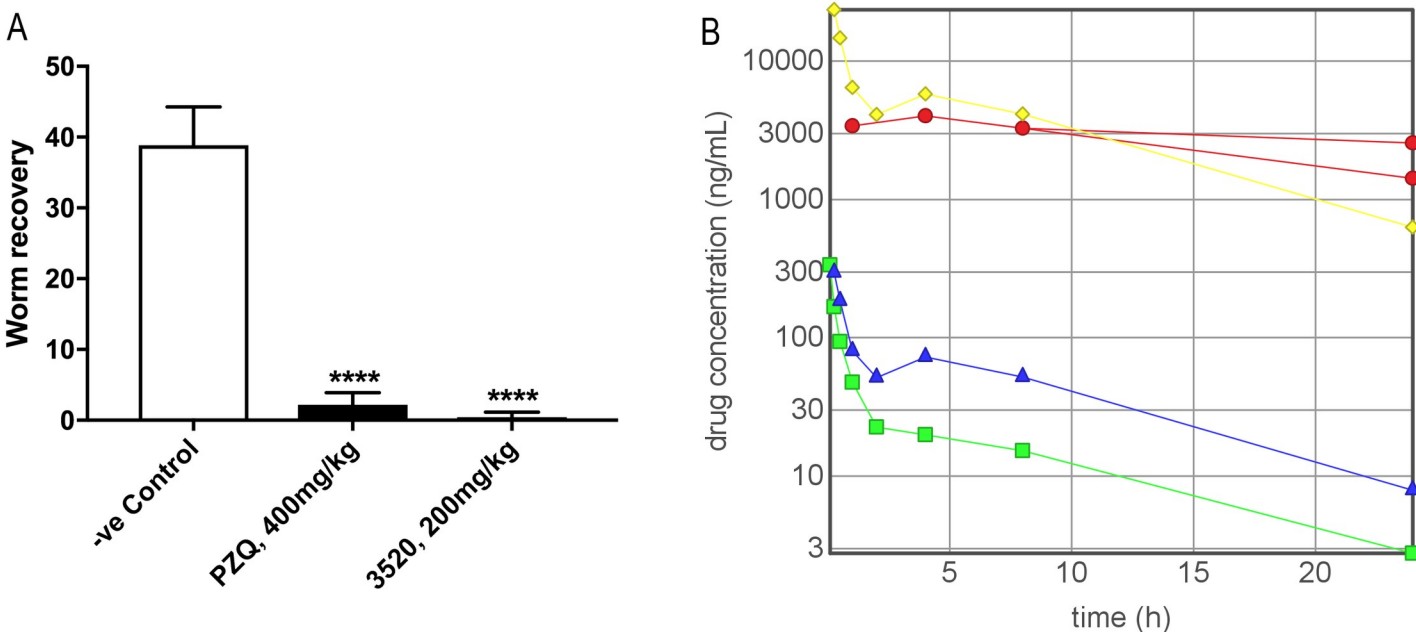

**Fig 6. A. *In vivo* efficacy against adult worms in the mouse model.** Graph shows the Mean (+standard deviation) worm recoveries following single oral dose treatment with vehicle or vehicle plus LSHTM-3520 (Aqueous formulation- see Materials and Methods) of 42-day old infections in mice (n = 6). **B. Drug concentration in plasma measured at different time points after administration.** Green rectangle—PK experiment, iv dose of LSHTM-3520 0.5 mg/kg co-administered with 4 other compounds, data points are mean values from 3 mice. Purple triangle—PK experiment, oral dose of LSHTM-3520 2.5 mg/kg co-administered with 4 other compounds, data points are mean values from 3 mice. Red circle—Efficacy experiment, oral dose of LSHTM-3520 200 mg/kg, data points are single values from five different mice each sampled once during the experiment (2 mice sampled at 24 h). Yellow diamond—Arithmetic scaling of the oral dose 2.5 mg/kg cassette dose experiment to simulate a dose of 200 mg/kg to show comparison with experimental measurements of 200 mg/kg oral dose.

there were no issues observed with any of the animals exposed to this level of drug (or indeed to the one mouse given a single dose of 400 mg/kg before the efficacy study was carried out).

## Improvements to reduce the predicted human dose

Our goal is to achieve a clinically effective treatment for schistosomiasis at a predicted dose of below 10 mg/kg. To make a human dose prediction several assumptions have been made.

- We assume that the plasma concentration achieved in mouse sufficient to kill worms is the same as the plasma concentration we need to achieve in humans to see the same effect. A component element is an assumption that the mechanism of *S. mansoni* worm reduction in mouse and human are the same.

- We assume that allometric scaling can be used to estimate human clearance based on mouse clearance. One factor that would make this assumption unsafe is if the clearance mechanisms and rates for our series in mouse and human were different. Fortunately, *in vivo* clearance correlates well with metabolic stability calculated from *in vitro* mouse hepatocyte measurements (Fig 7A). *In vitro* stability estimates in mouse and human hepatocytes are also fairly well correlated (Fig 7B). A third element of the assumption that mouse clearance can be used to predict human clearance is that protein binding occurs to the same degree in both species as is demonstrated in Fig 7C.

Estimates of dose prediction in human were made using the PK Tool [28]. A simple estimate of an effective human dose for LSHTM-3520 can be made by making the assumption that a

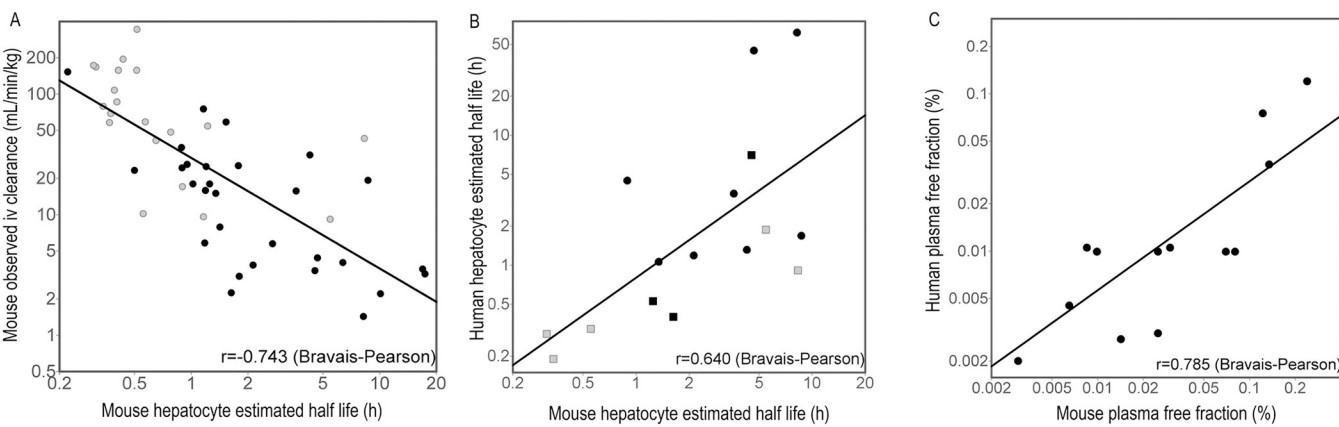

**Fig 7. Correlations between various mouse and human *in vitro* properties. A** *In vitro* half-life in mouse hepatocytes calculated from the percentage of test compound remaining after a set incubation time (30 minutes, grey circles or 90 minutes, black circles) compared with observed clearance measured in mouse after iv administration at 0.5 mg/kg co-administered in groups of five. **B** *In vitro* half-life in mouse or human hepatocytes calculated as described for panel A (incubation time in human hepatocytes 30 minutes, squares or 90 minutes, circles). **C** Plasma protein binding (expressed as free fraction), measured at 10uM. Despite difficulties in accurate measurement due to very high levels of protein binding, the data is consistent with equal levels of protein binding in mouse and human plasma.

plasma concentration of 2,000 ng/mL 24 h after oral administration (the same as that achieved in Fig 6B) would give sufficient clinical efficacy. Assuming human clearance can be predicted by allometric scaling from that in mouse (with an exponent of 0.75) and that other factors such as protein binding, blood/plasma ratio and bioavailability remain the same in human as in mouse, leads to a predicted single human oral dose of 48 mg/kg for LSHTM-3520. To design compounds with a lower predicted human dose we pursued two strategies, one to improve potency by optimisation of the core and substituents, the other to decrease clearance through blocking positions we thought likely to be vulnerable to metabolic enzymes and/or conformational restriction. Table 2 lists a selection of compounds synthesised following these strategies.

It was also crucial to our objectives to demonstrate *in vivo* efficacy against juvenile worms. The 4-fluoroindazole attached at the 5-position of LSHTM-3604 reduces clearance relative to the difluorophenol of LSHTM-3520. Fig 8 shows that both LSHTM-3520 and LSHTM-3604 are effective *in vivo* against juvenile worm infection with the significantly lower clearance of LSHTM-3604 resulting in efficacy at a lower dose than LSHTM-3520. For this experiment artemether was used as the positive control as PZQ is weak against juvenile worm infection.

These early experiments also allowed us to use the PK Tool [28] to predict the dose to use in the mouse efficacy models. As part of the prediction we need to recognise that the measured *in vitro* potency will be different from that in whole blood. As described earlier, the most potent compounds are lipophilic and quite rigid (although atropisomerism was not observed in $^1$H NMR experiments). They also display high protein binding in plasma, assay media and microsomal stability assays. *In vitro* potency assays were run using 10% of the level of serum proteins found in whole blood (in this case, Foetal Bovine Serum). Measurement of binding of LSHTM-3520 in the assay medium gave a free fraction, Fu of 0.0014. An experiment was run using LSHTM-3520 to determine whether the potency shift on varying the plasma protein level in the *in vitro* assay accords with prediction. Experiments run with 3%, 10% and 30% FBS resulted in $EC_{50}$ values of 8 nM, 24.4 nM and 75 nM respectively. This is in excellent agreement with theoretical expectation based on calculation, in which given high protein binding, a 3 fold increase in protein will decrease the free fraction 3 fold and result in a 3 fold larger $IC_{50}$ value. Since the *in vitro* $IC_{50}$ values are run at 10% of whole blood plasma protein, whole blood $IC_{50}$ values will be 10 fold higher numbers (ie weaker).

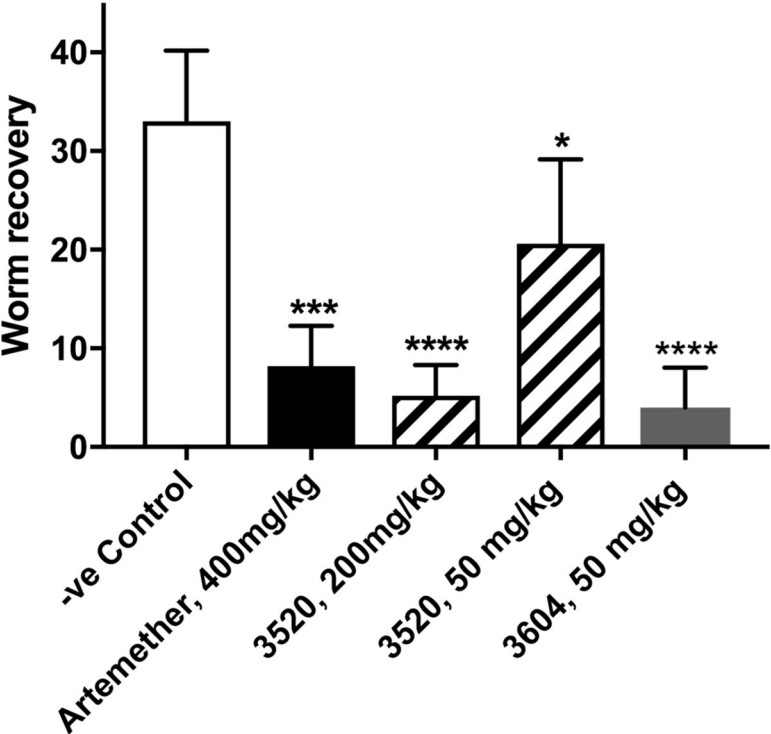

**Fig 8. *In vivo* efficacy against juvenile worms in the mouse model.** Graphs show the Mean (+ standard deviation) worm recoveries following single oral dose treatment with vehicle or vehicle plus drug (Aqueous formulation—see Materials and Methods) of 21-day old infections in mice (n = 6). Further details are in the Materials and Methods section.

Assuming linear kinetics the plasma levels scaled from the low-dose oral PK experiment (2.5 mg/kg) for LSHTM-3520 (8 ng/mL) after 24 h would be 160 ng/mL from 50 mg/mL and 640 ng/mL at 200 mg/mL. These plasma levels would be 1.3 and 5.3 fold above the calculated whole blood $IC_{50}$ value for LSHTM-3520. Using the same calculation, the plasma level after 24 h, scaled from 2.5 mg/kg dose for LSHTM-3604, would be 1,430 ng/mL which is 9.4 fold above the calculated whole blood $IC_{50}$ value of 152 ng/mL. This and further experiments allowed us to build up a body of evidence that efficacy in the mouse model was likely to be achieved when the predicted plasma concentration after 24 h was more than 5 fold higher than the calculated blood potency. It is not possible to determine whether AUC or plasma concentration at 24 h are more predictive as they are closely correlated but, given the small number of measurements made during the *in vivo* experiment and long half-life of the later compounds, the plasma concentration at 24 h is a convenient measure whereas we cannot so accurately measure $AUC_{inf}$ from 24 h sampling. What we can say with some certainty is that efficacy correlates better with plasma concentration at 24 hours than with plasma concentrations at shorter time points or with $C_{max}$. Human dose predictions were made using the PK Tool using the PKPD relationship that the predicted plasma concentration at 24 h should be 7 fold higher than the calculated adult worm blood potency.

Replacing the $^i$Pr of LSHTM-3604 (and LSHTM-3520) with trifluoromethyl at the 4-position of the imidazopyrazine gives LSHTM-3642 with an estimated half-life of 72 h in mouse, sampled over 24 h (33 h in rat, sampled over 4 days). In a head-to-head comparison between LSHTM-3642 and PZQ in the mouse model with adult worm infection (Fig 9A) LSHTM-3642 shows excellent *in vivo* efficacy from a single oral dose, with an $EC_{50} < 6.25$ mg/kg. Fig 9C

# Adult worm treatment

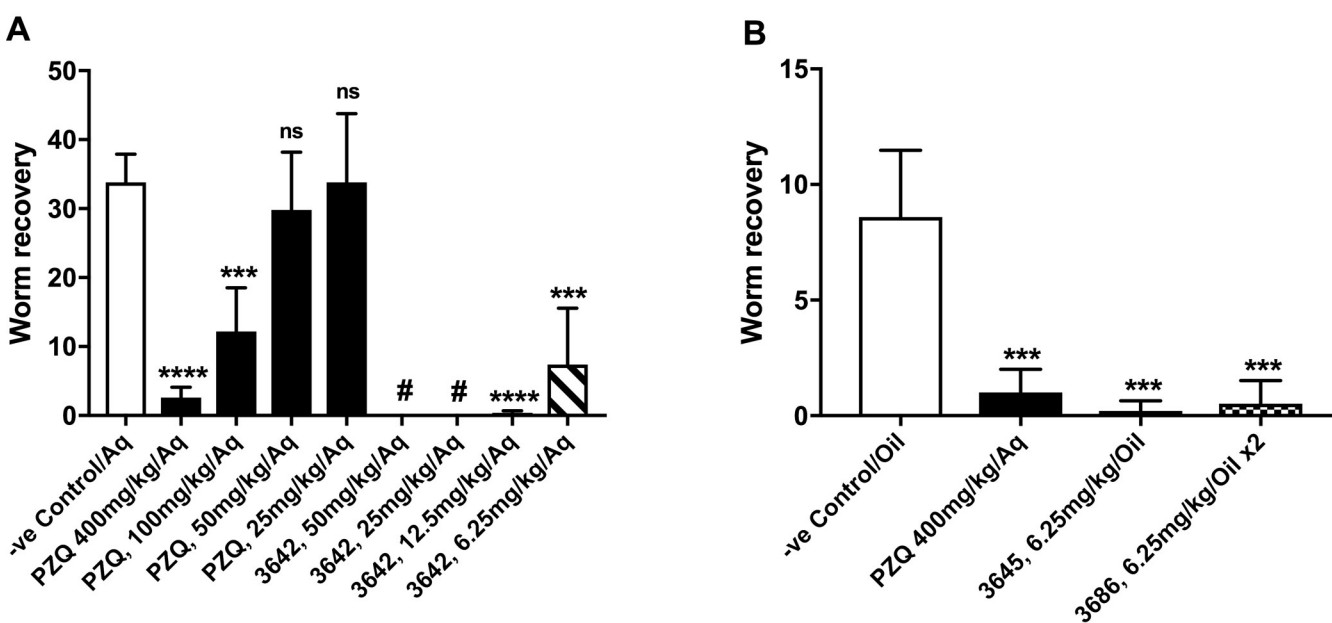

# Juvenile worm treatment

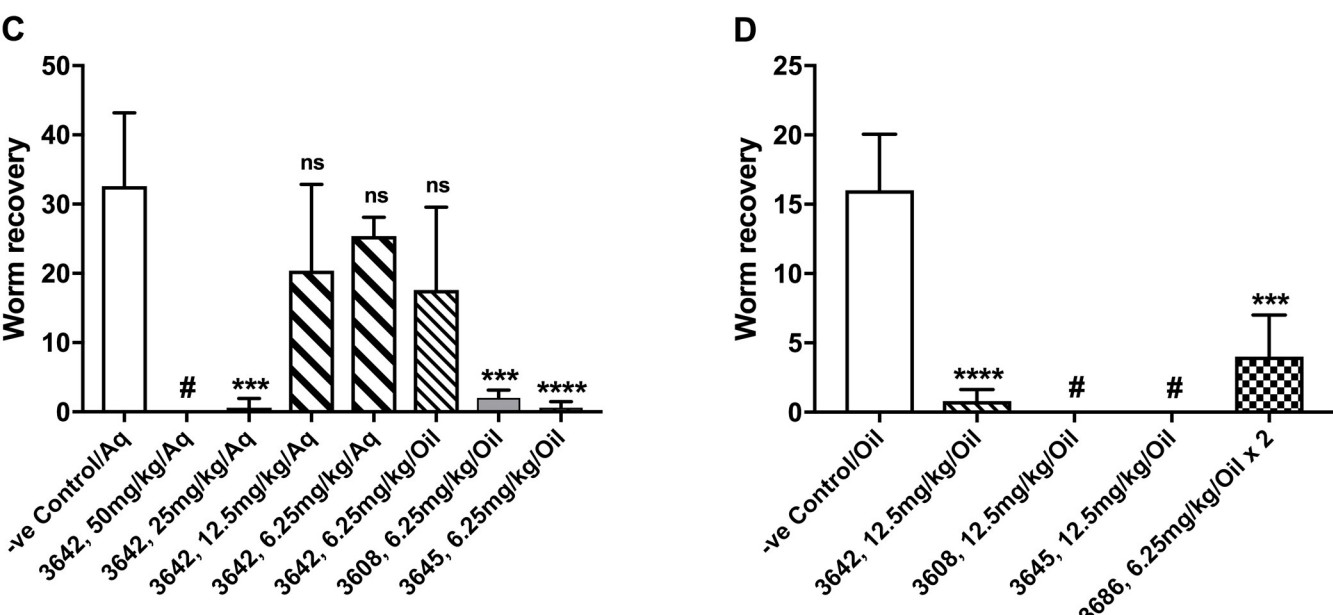

**Fig 9. *In vivo* demonstration of efficacy.** Graphs show the Mean (+ standard deviation) worm recoveries following single oral dose treatment with vehicle or vehicle plus drug of infections in mice (n = 5), #—no worms recovered (ie complete cure in all mice). The formulation of drug (or control) was either Aq (Drugs were suspended in 7% Tween-80 / 3% Ethanol / double distilled water and drug dispersal was facilitated by vortexing and using a sonicating water bath) or oil (Drugs were first dissolved in DMSO, then diluted with corn oil to give a 1:19 DMSO/corn oil ratio. **Panel A** shows a head to head comparison between PZQ and LSHTM-3642 *in vivo* against adult worms with a single oral dose giving PZQ an $EC_{50}$ between 50 and 100 mg/kg and LSHTM-3642 an $EC_{50}$ below 6.25 mg/kg. **Panel B** A demonstration of good efficacy from two compounds with low *in vitro* $EC_{50}$ values. LSHTM-3645 was used with a single oral dose, LSHTM-3686 was dosed twice, 12h apart because it has a shorter (6h) murine half-life than some of the other compounds shown. **Panel C** A dose response experiment showing that LSHTM-3642 has an $EC_{50}$ against juvenile worms *in vivo* of between 12.5 & 25 mg/kg. The experiment also shows that both

LSHTM-3608 and LSHTM-3645 have EC$_{50}$s below 6.25 mg/kg. **Panel D** This experiment shows that LSHTM-3645 and LSHTM-3608 are extremely effective in clearing juvenile worm infections *in vivo* from a 12.5 mg/kg oral single dose. As for panel B, LSHTM-3686 was dosed twice 12h apart because of its shorter half-life in mouse.

shows that LSHTM-3642 also has good efficacy in mouse against juvenile worms *in vivo*, with an EC$_{50}$ between 12.5 and 25 mg/kg.

Additional improvements in *in vitro* potency and metabolic stability could be obtained by larger polyfluorinated side chains, for example in LSHTM-3608 and LSHTM-3645. These changes resulted in compounds which display excellent efficacy against juvenile worms *in vivo* with LSHTM-3608 and LSHTM-3645 giving 100% and 98% worm reductions from single oral doses of 12.5 mg/kg and 6.25 mg/kg respectively (Fig 9C and 9D). LSHTM-3645 is similarly highly effective against adult worm infections (Fig 9B). The predicted human doses for LSHTM-3608 and LSHTM-3645 are 2.5 mg/kg and 3.5 mg/kg respectively. In addition to the cores described in Table 1, an additional core explored, pyrazolopyrimidine, is exemplified by compound LSHTM-3644 (Table 4, Fig 10). This compound carries the same substituents as compound LSHTM-3604, and indeed is only a little less potent *in vitro* (adult *S. mansoni* EC$_{50}$ 40 nM, juvenile *S. mansoni* EC$_{50}$ 60 nM). Since compounds with this core are both slightly weaker and show somewhat poorer plasma exposure they were not extensively investigated, although should some as yet unidentified issue with the imidazopyrazines occur it is possible the pyrazolopyrimidines may prove a viable alternative. Compounds LSHTM-3686 and LSHTM-3661 are even more potent *in vitro* against *S. mansoni* than LSHTM-3608 and LSHTM-3645, but they are less metabolically stable in hepatocytes (Table 2) and show higher clearance in mouse (Table 4). Due to the shorter half-life in mouse, we reasoned that higher or multiple doses would be required to demonstrate good worm reduction in the mouse model. Fig 9B and 9D indeed show that 6.25 mg/kg of LSHTM-3686 given twice 12 h apart induced significant worm burden reductions of 94% and 75% against adult or juvenile infections. However, the excellent *in vitro* potency coupled with good predicted human PK results in a dose prediction to human of 1 mg/kg from a single oral dose of LSHTM-3686, to cure humans of schistosomiasis.

Given the high potency of this series we evaluated some additional properties over the course of the project. LSHTM-3645 was tested against a panel of 38 *in vitro* toxicity targets and no reproduceable activity was observed at 1uM. LSHTM-3520, LSHTM-3604, LSHTM-3642 were tested against a panel of 50 human kinases at 1 μM with none showing greater than 25% inhibition. Neither LSHTM-3645 nor LSHTM-3686 showed any inhibition of the key ion channels hERG, Cav1.2, Kv1.5 and Nav1.5. No degradation of either compound was observed at 37˚C for 3 h in plasma, phosphate buffer (pH 8.0) or simulated gastric fluid (pH 1.2). Kinetic solubility data (Table 4) suggests that most compounds are poorly soluble at pH 7.4 but highly soluble in simulated fasted and fed gastric fluid.

## Discussion

This paper documents the discovery of a potent series of anti-schistosomiasis compounds with excellent prospects for achieving a single dose cure in humans that stand out from those reported in a recent extensive review of the field [17] and series reported since that review [29–31]. The weakly potent original hits have been optimised to compounds with approximately three orders of magnitude better *in vitro* potency. This has been achieved largely by judicious choice of polarity at R$_1$ and introduction of substitution at R$_4$. The window over cytotoxicity against human cell lines increased in step with potency improvement against the worms. The compounds were well tolerated in the efficacy studies and other *in vitro* assessments of

**Table 4. *in vitro*, physicochemical and *in vivo* properties of compounds pictured in Fig 10.**

| compound number | PK (mouse, 0.5 mg/kg iv, 2.5 mg mg/kg oral in a mixture of 5 compounds unless otherwise stated)[a] | | | | | | Efficacy experiments | | | | | | | Measured physicochemical properties | | | | | |
|---|---|---|---|---|---|---|---|---|---|---|---|---|---|---|---|---|---|---|---|
| | Clobs (L/h/kg) | half life (h) | AUCinf (h·ug/mL) | Vss (L/kg) | oral bioavailability (%) | Plasma conc @ 24 h after oral dose (ng/mL) | Age of worm infection | Oral Dose (mg/kg) | Formulation[d] | Worms remaining (%) | Plasma conc @24 h (ng/mL) | Observed plasma conc @24 h / calculated blood IC$_{50}$ | human single oral dose prediction (mg/kg) | Human plasma protein % binding (% recovery) | Mouse plasma protein % binding, % recovery) | Assay medium (%binding, % recovery) | Kinetic solubility, pH 7.4 (uM) | Kinetic solubility, FaSSIF (uM) | Kinetic solubility, FeSSIF (uM) |
| LSHTM-3520 | 1.08 | 7.0 | 0.5 | 7.1 | 50 | 8.1 | Adult | 200 | Aq | 1 | 1,907 | 9.2 | 25 | 99.9895 (97.1) | 99.97 (91.5) | 99.86 (69.3) | 12.7 | | |
| | | | | | | | Juvenile | 200 | Aq | 16 | 1,980 | 17 | | | | | | | |
| | | | | | | | Juvenile | 50 | Aq | 62 | 208 | 1.7 | | | | | | | |
| LSHTM-3604 | 0.26 | 10.5 | 1.9 | 3.4 | 60 | 71.6 | Juvenile | 25 | corn oil | 13 | 223 | 1.5 | 6 | 99.9973 (92.4) | 99.9858 (94.5) | | 0.834 | 200 | 200 |
| | | | | | | | Juvenile | 50 | Aq | 12 | 505 | 3.3 | | | | | | | |
| LSHTM-3642 | 0.13 / 0.20[b] | 74 / 33[b] | 5.5 / 5.0[b] | 8.7 / 8.8[b] | 83 / 67[b] | 227 / 357[b] | Adult | 12.5 | Aq | 1 | 939 | 7.1 | 20 | 99.9895 (97.4) | 99.9915 (92.1) | | 0.854 | 196 | 197 |
| | | | | | | | Juvenile | 25 | Aq | 1.8 | 1,700 | 7.4 | | | | | | | |
| | | | | | | | Juvenile | 12.5 | corn oil | 5 | 1,320 | 5.7 | | | | | | | |
| LSHTM-3608 | 0.09 | 37.1 | 6.2 | 4.0 | 100[c] | 378 | Juvenile | 12.5 | corn oil | 0 | 879 | 8.1 | 2.5 | 100 (97.8) | 99.9195 (80.8) | | 0.704 | 200 | 196 |
| LSHTM-3645 | 0.21 | 23.4 | 2.4 | 6.4 | 100[c] | 241 | Adult | 6.25 | corn oil | 1 | 1,012 | 19.6 | 3.5 | 100 (85.8) | 100 (96.9) | | 0.264 | 199 | 200 |
| | | | | | | | Juvenile | 6.25 | corn oil | 1.8 | 708 | 12.4 | | | | | | | |
| | | | | | | | Juvenile | 12.5 | corn oil | 0 | 984 | 17.3 | | | | | | | |
| LSHTM-3686 | 0.94 | 6.2 | 0.5 | 3.9 | 100[c] | 16 | Adult | 6.25x2 (12 h apart) | corn oil | 2.4 | 241 | 10.4 | 1 | 99.998 (62.7) | 99.997 (76.1) | | 0.885 | 33.9 | 43.8 |
| | | | | | | | Juvenile | 6.25x2 (12 h apart) | corn oil | 25 | 37.2 | 2.1 | | | | | | | |
| LSHTM-3661 | 1.88 | 4.5 | 0.3 | 5.6 | 57.1 | 1.2 | Juvenile | 50 | corn oil | 0 | 178 | 14.4 | 3 | | | | 0.04 | | |
| LSHTM-3690 | 1.47 | 4.6 | 0.3 | 5.0 | 100[c] | 7.4 | | | | | | | 5 | 99.925 (89) | 99.8775 (93.9) | | | 188 | 196 |
| LSHTM-3644 | 0.38 | 6.3 | 1.33 | 2.4 | 76 | 35 | | | | | | | | | | | | | |

(a) PK values are from dosing at 0.5 mg/kg iv (Clobs, half-life, AUC, Vss,) and 2.5 mg/kg oral (bioavailability, plasma concentration at 24 h). (b) rat, iv 1 mg/kg, po 10 mg/kg, single compound. (c) AUC$_{inf}$ is subject to significant error for these very long half-life compounds as samples were only taken up to 24 h. In some cases this results in the calculation of bioavailability resulting in a value > 100%. If so, this is shown in Table 4 as 100. (d) See Materials and Methods section for formulation details.

**Fig 10. Structures of compounds shown in Table 4.**

potential toxicity concerns such as ion channel block and inhibition of a set of human safety targets were all clean.

The mechanism of action of this series is currently unknown. This is a common situation for anthelmintics; in fact the mechanism of PZQ is still a matter of some debate [32]. The structure of our compounds and relationship to the previously disclosed anti-malarial series exemplified by compound compound 11 (Table 1, [26] encouraged us to test examples against a panel of human kinases. The lack of activity against the examples tested does not rule out *Schistosoma* kinase inhibition as a mechanism but does suggest that if that is the mechanism of action, the compounds are not promiscuous pan-kinase inhibitors.

Compounds such as LSHTM-3645 and LSHTM-3608 have very long half-lives in mouse which along with good potency against both juvenile and adult *S. mansoni* allow them to clear worms from mice at very low doses. We believe that the excellent metabolic stability of many of the later compounds is conferred by a combination of substituents (eg fluorination) which are known to block cytochrome P450 metabolism, and conformational rigidity. These compounds are predicted to have extremely long half-lives in human. This property could be useful in the clinic, for example in mass drug administration or prophylaxis. However, very long half-lives can make clinical development challenging. Fortunately the series also affords compounds (such as LSHTM-3686, LSHTM-3661, LSHTM-3690) with similar human dose predictions of between 1 and 5 mg/kg from a single oral dose with shorter half-lives.

The key next step in development is to assess safety *in vivo*. The series displayed poor solubility at pH 7.4 but with high permeability and might be classified as BCS (Biopharmaceutics Classification System) II. In order to assess safety, it will be necessary to find formulations that allow for high multiples of the predicted efficacious dose to be administered in suitable animal species. Hopefully methods used for safety studies of other BCS class II drugs will be applicable in this series. However, dose formulation at or around the predicted human dose was straightforward (for example in the PK studies) and consequently it is anticipated that a fairly simple formulation could ultimately be used in the clinic.

In summary, the series offers versatility in the selection of a candidate which could be selected predominantly for treatment of infection (with a relatively short wash out time expediting safety and clinical trials) through to candidates with half-lives that may make them suitable for prophylaxis. In addition, so far the series shows an excellent *in vitro* and *in vivo* safety profile and can be made in relatively few steps. In comparison to praziquantel the series is

more potent, with longer half-life and, in contrast to praziquantel, shows good activity against both juvenile and adult worms. Several compounds are predicted to effect a single dose cure for juvenile and adult schistosome infections with a dose of 1–5 mg/kg and thus offers a compelling opportunity to develop a drug for this debilitating disease.

## Supporting information

**S1 Text. Synthesis of LSHTM-3642.** A detailed description of the synthesis and characterisation of LSHTM-3642 and the intermediates leading to it from commercial starting materials.
(DOCX)

**S1 Table. Cross reference to patent.** A look-up table between compounds in this paper and compounds in patent WO2018130853 [21] where the synthesis and characterisation of further examples is described.
(DOCX)

## Acknowledgments

This paper is dedicated to the memory of Dr Tanya Parkinson, without whom the project would never have happened. The authors would like to thank a large number of people who have helped the project along the way. In particular, Dr Tim Wells (MMV) provided invaluable guidance and input across a large range of topics throughout the project, provided the compound library which gave us the hits that were optimised and funded the provision of bulk material to help ensure speedy progression into development. Prof. Paul Fish (UCL), Prof. Dennis Smith (consultant), Dr Alan Brown (Salvensis) and Prof Simon Croft (LSHTM) all gave invaluable support and advice. Dr Thomas Spangenberg (Merck KGaA) also followed the project with interest; Merck have now taken the project on and are aiming to conduct experiments prior to development [33] Dr Oliver Kingsbury (Elkington & Fife) provided enormous help and time for free in preparation of the patents. We would also like to thank Sophie Howson and Rachel Gregory for assistance with maintenance of the *S. mansoni* life cycle at LSHTM, BRI-NIH, USA for providing us with infected *Bulinus* and *Oncomelania* snails, Dr Susanta Mondal, Dr Bikash Maity, Dr Mrinal Kundu & their colleagues at TCG Life Sciences and Jill Segelbacher at York Bioanalytical. Data storage, analysis and visualisations were carried out with ScienceCloud, KNIME, Prism and DataWarrior.

## Author Contributions

**Conceptualization:** J. Mark F. Gardner, Nuha R. Mansour, Andrew S. Bell, Quentin Bickle.

**Data curation:** J. Mark F. Gardner, Nuha R. Mansour, Andrew S. Bell, Quentin Bickle.

**Formal analysis:** J. Mark F. Gardner, Nuha R. Mansour, Andrew S. Bell, Quentin Bickle.

**Funding acquisition:** J. Mark F. Gardner, Quentin Bickle.

**Investigation:** Nuha R. Mansour, Helena Helmby, Quentin Bickle.

**Methodology:** J. Mark F. Gardner, Nuha R. Mansour, Andrew S. Bell, Quentin Bickle.

**Project administration:** J. Mark F. Gardner, Nuha R. Mansour, Quentin Bickle.

**Resources:** J. Mark F. Gardner, Nuha R. Mansour, Quentin Bickle.

**Software:** J. Mark F. Gardner.

**Supervision:** J. Mark F. Gardner, Nuha R. Mansour, Andrew S. Bell, Quentin Bickle.

**Validation:** J. Mark F. Gardner, Nuha R. Mansour, Andrew S. Bell, Helena Helmby, Quentin Bickle.

**Visualization:** J. Mark F. Gardner, Nuha R. Mansour, Quentin Bickle.

**Writing – original draft:** J. Mark F. Gardner, Nuha R. Mansour, Andrew S. Bell, Quentin Bickle.

**Writing – review & editing:** J. Mark F. Gardner, Nuha R. Mansour, Andrew S. Bell, Helena Helmby, Quentin Bickle.

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
