## [Decision Letter · Decision Letter 0]

16 Mar 2021

Dear Dr Gardner,

Thank you very much for submitting your manuscript "The discovery of a novel series of compounds with single-dose efficacy against juvenile and adult Schistosoma species" for consideration at PLOS Neglected Tropical Diseases. As with all papers reviewed by the journal, your manuscript was reviewed by members of the editorial board and by several independent reviewers. The reviewers appreciated the attention to an important topic. Based on the reviews, we are likely to accept this manuscript for publication, providing that you modify the manuscript according to the review recommendations. 

I agree with all the reviewers that this manuscript represents an important contribution that identifies new compounds with therapeutic potential against schistosomes. I also agree with some of the concerns of the reviewers. The authors might consider the suggestion of Reviewer 1 to split off the medicinal chemistry section, though I would not consider that a requirement for acceptance. Most of the other concerns were minor. However, if resubmitting, I would strongly urge the authors to include page and line numbers to make reviewing less cumbersome.

Sincerely,

Robert M Greenberg

Associate Editor

Jennifer Keiser

Deputy Editor

I agree with all the reviewers that this manuscript represents an important contribution that identifies new compounds with therapeutic potential against schistosomes. I also agree with some of the concerns of the reviewers. The authors might consider the suggestion of Reviewer 1 to split off the medicinal chemistry section, though I would not consider that a requirement for acceptance. Most of the other concerns were minor. However, if resubmitting, I would strongly urge the authors to include page and line numbers to make reviewing less cumbersome.

Reviewer's Responses to Questions

**Key Review Criteria Required for Acceptance?**

**Methods**

-Are the objectives of the study clearly articulated with a clear testable hypothesis stated?

-Is the study design appropriate to address the stated objectives?

-Is the population clearly described and appropriate for the hypothesis being tested?

-Is the sample size sufficient to ensure adequate power to address the hypothesis being tested?

-Were correct statistical analysis used to support conclusions?

-Are there concerns about ethical or regulatory requirements being met?

Reviewer #1: The objectives are clearly described and the design is appropriate. My concern about the methods represents a weakness in the manuscript, which contains a significant component of medicinal chemistry, a feature not usually found in papers published in this journal. Methods for the synthesis of the compounds evaluated in bioassays are typically provided in detail; this is not the case here, and the authors refer to the contract lab that made them. This needs to be addressed. Indeed, although this is a very important manuscript, I believe it should be split into two (a rare recommendation, to be sure): the med them work would be better placed in a more relevant journal, with details of the syntheses and perhaps the juvenile in vitro bioassay data as the biological component. The manuscript for PLoS-NTDs should focus only on the results of the most promising compounds (and these results are truly exciting).

Reviewer #2: It would be helpful to have a fuller description of the methods used in some instances, rather than just a reference.

Reviewer #3: well done

**Results**

-Does the analysis presented match the analysis plan?

-Are the results clearly and completely presented?

-Are the figures (Tables, Images) of sufficient quality for clarity?

Reviewer #1: I have no particular concerns about these matters.

Reviewer #2: Yes

Reviewer #3: yes

**Conclusions**

-Are the conclusions supported by the data presented?

-Are the limitations of analysis clearly described?

-Do the authors discuss how these data can be helpful to advance our understanding of the topic under study?

-Is public health relevance addressed?

Reviewer #1: The data presented justify the conclusion that the authors have identified compounds that warrant pre-clinical (toxicology) and clinical development as candidates for the treatment of schistosomiasis. I congratulate them on their progress.

Reviewer #2: Yes

Reviewer #3: yes

**Editorial and Data Presentation Modifications?**

Reviewer #1: The lack of page and line numbering is an impediment for me, but I offer a few minor suggestions for improvement.

1. Abstract: although PZQ is given in a high dose, the cost of goods is low enough to permit it. Rather than refer to the size of the dose, the authors should prioritize ease of administration with a cos-of-goods compatible with eventual donation.

2.Throughout the manuscript, separate numbers from units (i.e., 4 mM instead of 4mM, etc.)

3.Last sentence of the Introduction: these are candidates for clinical development, not fir use

4. M&M: were the observers of the mice blinded to the treatment? What kinds of adverse events were they trained to look for?

5. 1st sentence, Bioanalysis of plasma samples: the meaning of "all samples were extracted using protein precipitation" is unclear; please provide missing details.

6. Results, 2nd paragraph, 7: did the imidazopyrazine actually confer greater activity, or only appear to?

7. Last sentence, legend to Fig. 5:"overall it looks as though binding is similar between mouse and human plasma"; is the binding similar or is it not?

8. First sentence after this legend: use 'human' instead of 'man'. Your meaning is clear, but this usage is now outdated

9. Last sentence in the 2nd paragraph before Fig. 7: replace 'infected by' with 'of'

10. Next sentence after this: replace 'low dose efficacy' with 'high potency'

11. I could find no mention of the adoption of this project by Merck KGaA in the citation (33) provided in the Acknowledgments. Please update/clarify.

Reviewer #2: P21 - … two most clinically relevant species S. mansoni and S. haematobium …

Testing against both adult and larval worms of all three species would be informative.

P34, Table 4, formulation column ‘F2’ an ‘F1’ should be replaced by appropriate ‘Aq’ or ‘corn oil’

Reviewer #3: (Very) Minor Revision, see below

**Summary and General Comments**

Reviewer #1: As noted, this is a very important advance in the field and I think it should be punished in PLoS-NTDs. However, it is somewhat loosely written, and the combination of extensive medicinal chemistry with the highly topical data on in vivo activity is somewhat awkward. I repeat that I think the authors would be better served by separating these components into two distinct manuscripts.

Reviewer #2: Manuscript PNTD-D-21-00141 presents compelling and exciting results on the development of compounds efficacious for schistosomiasis therapy. The manuscript is well written and concise.

Reviewer #3: The paper of Gardner et al. deals with an important topic schistosome research, finding new compounds with efficacy against juvenile and adult worms.

The authors report on an approach optimising compounds discovered by high throughput screenings of compounds for clinical development. Their best hits showed clearance of juvenile and adult worms in a mouse model with a single oral dose < 10 mg/kg. Further compounds were predicted to be useful for treating schistosomiasis in humans with a single oral dose of < 5 mg/kg.

In their detailed results, the authors present a series of molecules with promising in vitro and in vivo safety profiles as well as with pharmacokinetic data demonstrating a longer half-life compared to praziquantel and reasonable activity against both juvenile and adult worms. Several compounds may have the potential to continue development to a single-dose cure for schistosomiasis. 

The article is a nice progress for the field and will be of interest to the audience of PLoS NTD. I have only a small number of minor comments:

- M&M, Pharmacokinetics: replace 4000 rpm by 4,000 rpm. Add space between 30 and µl. 

- Here and elsewhere, spaces between numbers and units are often missing, example: 10 μL of 10mM (in: Plasma Protein Binding in mouse or human plasma …), or pH7.4 on page 30.

- Use italics for species names throughout the text (see e.g. S. mansoni in the legend of Table 1).

PLOS authors have the option to publish the peer review history of their article (what does this mean?). If published, this will include your full peer review and any attached files.

Reviewer #1: No

Reviewer #2: No

Reviewer #3: No

Figure Files:

Data Requirements:

Reproducibility:

References

---

## [Editor Report · Decision Letter 1]

23 May 2021

Dear Dr Gardner,

We are pleased to inform you that your manuscript 'The discovery of a novel series of compounds with single-dose efficacy against juvenile and adult Schistosoma species' has been provisionally accepted for publication in PLOS Neglected Tropical Diseases.

Best regards,

Robert M Greenberg

Associate Editor

Jennifer Keiser

Deputy Editor

---

## [Editor Report · Acceptance letter]

9 Jul 2021

Dear Dr Gardner,

We are delighted to inform you that your manuscript, "The discovery of a novel series of compounds with single-dose efficacy against juvenile and adult Schistosoma species," has been formally accepted for publication in PLOS Neglected Tropical Diseases.

Best regards,

Shaden Kamhawi

co-Editor-in-Chief

Paul Brindley

co-Editor-in-Chief
